# A game theoretic approach identifies conditions that foster vaccine-rich to vaccine-poor country donation of surplus vaccines

Adam Lampert[1✉], Raanan Sulitzeanu-Kenan[2✉], Pieter Vanhuysse[3] & Markus Tepe[4]

## Abstract

**Background** Scarcity in supply of COVID-19 vaccines and severe international inequality in their allocation present formidable challenges. These circumstances stress the importance of identifying the conditions under which self-interested vaccine-rich countries will voluntarily donate their surplus vaccines to vaccine-poor countries.

**Methods** We develop a game-theoretical approach to identify the vaccine donation strategy that is optimal for the vaccine-rich countries as a whole; and to determine whether the optimal strategy is stable (Nash equilibrium or self-enforcing agreement). We examine how the results depend on the following parameters: the fraction of the global unvaccinated population potentially covered if all vaccine-rich countries donate their entire surpluses; the expected emergence rate of variants of concern (VOC); and the relative cost of a new VOC outbreak that is unavoidable despite having surplus doses.

**Results** We show that full or partial donations of the surplus stock are optimal in certain parameter ranges. Notably, full surplus donation is optimal if the global amount of surplus vaccines is sufficiently large. Within a more restrictive parameter region, these optimal strategies are also stable.

**Conclusions** Our results imply that, under certain conditions, coordination between vaccine-rich countries can lead to significant surplus donations even by strictly self-interested countries. However, if the global amount that countries can donate is small, we expect no contribution from self-interested countries. The results provide guidance to policy makers in identifying the circumstances in which coordination efforts for vaccine donation are likely to be most effective.

## Plain language summary

In an unequal world with open economies, pandemics do not stop at national borders. Higher-income countries may then benefit from helping lower-income countries. In particular, since new variants of a virus may emerge in vaccine-poor countries, vaccine-rich countries may have a strong incentive to donate their surplus vaccine doses rather than stocking these domestically. But under which conditions will such self-interested donations occur? We develop a game-theoretic model, used to analyze the strategies of rational decision makers and how they depend on strategies of other decision-makers. We show that, if vaccine-rich countries can vaccinate a large share of the vaccine-poor world, it is optimal for them to donate their surplus vaccines. In certain circumstances, such donations are also a stable solution. These findings may inform health policy on ways of improving the effectiveness of coordinated international vaccine donations.

[1] Institute of Environmental Sciences, Robert H. Smith Faculty of Agriculture, Food and Environment, The Hebrew University of Jerusalem, Rehovot, Israel. [2] Federmann School of Public Policy, The Hebrew University, Jerusalem, Israel. [3] Department of Political Science and Danish Institute for Advanced Study, University of Southern Denmark, Odense, Denmark. [4] Institute of Social Sciences, University of Oldenburg, Oldenburg, Germany. ✉email: adam.lampert@mail.huji.ac.il; raanan.s-k@mail.huji.ac.il

The COVID-19 pandemic has shone a harsh light on the urgent need to better understand the drivers of policy solutions to global infectious disease crises. By November 2021, the pandemic had claimed the lives of more than 5.23 million people worldwide[1], and had led to severe economic losses. During most of 2020 and before effective vaccines and treatments were available, containing the spread of the virus compelled countries to adopt lockdowns and social distancing policies. In a remarkable scientific achievement, safe and effective vaccines were developed in record time[2,3]. The availability of COVID-19 vaccines offers countries and the international community important new means to combat the pandemic. At this particular stage in pandemic cycles, the key questions are how best to utilize the available vaccines on a global scale, and what strategies are likely to receive the required international cooperation for their implementation.

Scarcity in supply, coupled with unequal allocation of the available vaccines across countries, present the most formidable challenges for maximizing the potential benefits of these vaccines[4]. Allocating doses internationally in proportion to countries' population sizes is estimated to be a close-to-optimal strategy in terms of averting deaths worldwide[5] and benefiting the economy, both globally and in donor countries[6,7]. Nevertheless, extremely unequal vaccine distribution has typified the availability of vaccines across countries[2,8]. This situation means that most people in the world's poorest countries might not have access to COVID-19 vaccines until at least mid-2023[8]. Since about 85% of the global population resides in low- and middle-income countries, most of humanity remains exposed to continued outbreaks[8]. This situation increases the risk that further virus variants will emerge, possibly undermining the efficacy of existing vaccines[2,3].

A notable attempt to distribute vaccines from high-income to low-income nations is COVAX - the COVID-19 Vaccine Global Access Facility. Led by the World Health Organization and coordinated jointly with the Coalition for Epidemic Preparedness Innovations and the Global Alliance for Vaccines and Immunization, COVAX is a pooled procurement initiative that aims to provide all countries with COVID-19 vaccines at differential prices[2,3,9]. The guiding principle of COVAX was to prioritize vaccination globally by sub-populations: from older adults, healthcare workers, and other high-risk individuals to the wider sections of the population. It stipulates that no country should vaccinate more than 20% of its population until all countries have vaccinated 20% of their populations[3].

However, the COVAX initiative has so far failed to come close to its objectives[10], due to the aggregate consequences of vaccine nationalism[11]. Many of the world's wealthiest countries have adopted procurement strategies that prioritize widespread inoculation of national populations ahead of the vaccination of health care workers and high-risk populations in low-income countries[2,3].

The challenges facing a globally effective supply of vaccines through supranational initiatives such as COVAX have led to initiatives that focus on country-level and ad hoc intergovernmental approaches, such as donations by high-income countries of some of their pre-purchased vaccines to middle and low-income countries[12]. The G7 summit of June 2021 jointly pledged to provide 1 billion COVID-19 vaccine doses until June 2022[9]. Such initiatives focus on ways to redistribute the excess stocks of doses accumulated by vaccine-rich countries *after* they have vaccinated large shares of their own populations. They echo similar attempts in previous global health crises, such as smallpox in the 1970s[13], HIV in the 1980s[2], and H1N1 in 2009[3]. This study adopts a game-theoretical approach to identify the conditions under which self-interested vaccine-rich countries

would donate their surplus vaccine doses (beyond those needed for the initial vaccination of their own populations) to vaccine-poor nations, rather than stocking these surplus doses domestically for their own future use. (Note that, to avoid expiration of stored vaccine vials, stocking in practice might be based on pre-purchased doses that can be supplied on short notice from the manufacturer[14]. Vaccine-rich countries have at least two self-interested reasons for sharing their vaccine surpluses with vaccine-poor countries. First, they have large open economies that are dependent on international trade and thus require significant levels of international travel and open borders. Second, their efforts to stop the pandemic within national borders could be undermined by the emergence of new variants of concern (VOCs). Such a VOC (e.g., Delta and Omicron variants of COVID-19) may require increasing the level of immunization in the population by administering a booster[15]. In such circumstances, having a surplus stock can significantly shorten time-to-delivery, thereby better containing the outbreak. The probability of a VOC is a crucial factor in understanding vaccine-rich governments' willingness to re-allocate vaccines to vaccine-poor countries. The fear of VOC can either increase short-term national self-interest and a tendency to reserve health resources for domestic purposes or create awareness that long-term pandemic control can only succeed by effective global vaccination. Another complicating element is that donation by one country benefits all others by reducing the probability of future variants of the virus and their consequent costs; however, such a donation incurs a cost to the donating country by leaving it without available surplus in the case of a VOC outbreak.

Previous game-theoretical models applied to the challenge of epidemics, notably the "vaccination game"[16,17], have addressed individual-level decisions to vaccinate or not, assuming the availability of vaccines for the studied population. The current study points to the significance of international vaccine inequality in the context of a pandemic (defined as "an epidemic occurring worldwide or over a very wide area, crossing international boundaries"[18]), notably the fact that in many countries, vaccines are in short supply or practically unavailable, while other countries possess surplus stocks. These surplus stocks are controlled by the governments of vaccine-rich countries, and these country-level decisions are the focus of our model.

This study models (1) the vaccine donation strategy that is optimal for all vaccine-rich countries combined (a social planner's perspective); and (2) whether the optimal solutions could be adopted by the relevant countries, assuming strictly self-interested motivation on their part. Given that vaccine donations are dependent on the choices of vaccine-rich countries, the analysis strictly takes these countries' perspectives – both as a collective and as independent actors. In particular, we examine how the answers to these questions depend on key pandemic parameters: (1) the fraction of the global unvaccinated population potentially covered if all vaccine-rich countries fully donate their surplus ($v_{max}$); (2) the baseline expected annual rate of VOCs ($\lambda$); and (3) the fraction of the total cost of a new VOC outbreak that is unavoidable despite having surplus doses ($\alpha$). The game-theoretical model we develop is general in the sense that it identifies the conditions under which a minority of vaccine-rich countries is likely to donate a costly remedy in the context of a pandemic. By considering the realistic ranges of the model parameters in the case of COVID-19, we can cautiously infer more specific implications for potential international cooperation in coping with this particular pandemic.

## Methods

**The vaccine donation game**. We consider the strategies of $N$ vaccine-rich countries (hereafter: "the countries"). Beyond the

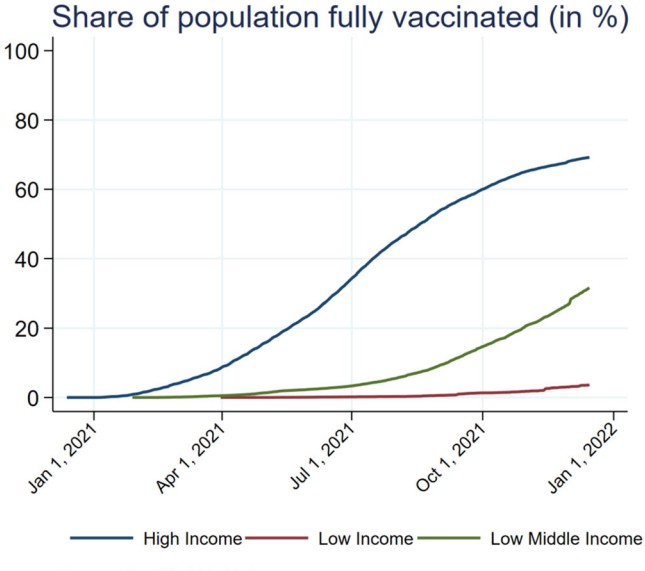

### Share of population fully vaccinated (in %)

Source: Our World in Data

— High Income  — Low Income  — Low Middle Income

**Fig. 1 The share of population fully vaccinated.** The share of the population fully vaccinated against COVID-19 across high-, lower middle-, and low-income countries (source: Our World in Data, https://github.com/owid/covid-19-data/tree/master/public/data).

vaccine doses needed for vaccinating their own entire population, each vaccine-rich country is assumed to have purchased additional stock sufficient to vaccinate its population two *additional* times. We assume that the benefit of having this surplus full-population stock is that it could be administered in case a VOC emerges. Each country can decide how much of the two extra doses per capita to donate to vaccine-poor countries: $s_i = 0, 1,$ or 2, where $i = 1, 2, 3, \ldots N$ denotes the country. Each country can donate zero extra doses and thus keep two extra doses per capita for itself; donate one dose and keep 1 for itself; or donate both extra doses and keep none (1 dose may protect against 1 variant; 2 doses may protect against 2 variants).

We denote $v$ as the fraction of the global unvaccinated population that could be vaccinated due to vaccine-rich countries' donations. Specifically, assuming that countries will donate only after fully vaccinating their domestic population (see Fig. 1 as a motivational illustration justifying our assumptions), "global unvaccinated population" refers to the unvaccinated in vaccine-poor countries. $v$ could vary from zero if no country donates ($s_i = 0$ for all $i$) to $v_{max}$ if all vaccine-rich countries donate both doses ($s_i = 2$ for all $i$). Specifically, if $v_{max} \geq 1$, the countries could vaccinate the entire unvaccinated world population and be fully protected against any variants. However, if $v_{max} < 1$, there is still a chance that variants will emerge even if all vaccine-rich countries contribute all their extra doses.

In turn, we denote $\lambda$ as the expected number of VOCs that emerge in unvaccinated populations within one year if no doses are donated. We assume that variants can emerge independently with some small probability in each unvaccinated person[19]. Therefore, if vaccines are donated, the expected annual rate of VOCs is given by $\lambda(1 - v)$ if $v \leq 1$ and 0 if $v > 1$. Also, we assume that variants occur independently of one another: The probability of a variant to emerge does not depend on the number of other variants that emerge. Equivalently, the emergence of a variant does not change the probability that other variants emerge. Therefore, the probability distribution of the number of variants is given by the Poisson distribution[20]. Specifically, if $v < 1$, the probability distribution of the number of VOCs that emerge is

given by a Poisson distribution with a mean $\lambda(1 - v)$. It follows that the probability for the occurrence of exactly $k$ variants, $P_k$, is given by the Poisson coefficient

$$P_k = \frac{1}{k!} [\lambda(1 - v)]^k e^{-\lambda(1-v)}. \tag{1}$$

Donation by country $i$ benefits all countries because it reduces the probability of future virus variants and their consequent costs. However, a donation also comes with a cost to the donor country because it might be left without an effective vaccine if variants do occur. Specifically, if a country has stocked enough doses to cover the occurrence of a variant, it will have to bear only a fraction $\alpha$ ($0 < \alpha \leq 1$) of the total cost of the corresponding outbreak. Note that $\alpha$ is expected to be greater than zero – i.e., a certain cost of a future outbreak is practically unavoidable, since some time is required for administering surplus vaccines to the population. For example, if a booster shot is required, the time and direct costs of administering the booster shot remain. Note that $\alpha$ may, more generally, capture the time and costs required to apply any stocked resource. For simplicity, we assume that if there is more than one outbreak, the cost $\alpha$ is the same in the first and the second outbreaks. During this interim period, the country is expected to bear the cost of the variant outbreak. Therefore, the expected cost of future outbreaks to country $i$ is given by

$$C_i = \beta P_{\geq 3} + \beta \cdot \begin{cases} \alpha P_1 + 2\alpha P_2 & \text{if } s_i = 0 \\ \alpha P_1 + P_2 & \text{if } s_i = 1 \\ P_1 + P_2 & \text{if } s_i = 2 \end{cases} \tag{2}$$

where $\beta$ is the cost due to an outbreak if the country does not have extra vaccine doses, and $P_{\geq 3}$ is the probability that three or more variants emerge. Note that $P_1$, $P_2$ and $P_{\geq 3}$ depend on $v$ (Eq. 1), therefore they depend on $s_i$ and on the strategies of all other countries.

**Analysis of the model: Optimal and stable solutions.** Using this model, we address the following two tasks. First, we identify the optimal solutions from the point of view of a 'vaccine-rich-world' social planner who aims to maximize the social welfare of all vaccine-rich countries combined. Specifically, we assume risk-neutral countries, namely, the optimal solution is given by the choice of $s_i$ for each $i$ that minimizes the total expected cost of outbreaks to all vaccine-rich countries:

$$\text{minimize} \sum_{i=1}^{N} C_i \tag{3}$$

After identifying the optimal solution, we ask whether this solution is stable. Assuming that each vaccine-rich country chooses the strategy $s_i$ that minimizes its own expected cost, will the optimal solution be adopted? To answer this question, we consider two solution concepts. The first concept is the Nash equilibrium. Assume, for example, that $s_i = 2$ for each country is the optimal solution. This is also a Nash equilibrium if and only if it does not benefit any of the countries to unilaterally change its strategy to $s_i = 1$ or $s_i = 0$. However, the Nash equilibrium solution concept is somewhat restrictive, because it does not take into account the responses of other countries to the deviation of a given country.

Another widely used solution concept in the context of international cooperation is a self-enforcing international agreement (henceforth SEA)[21]. The assumption is that each country chooses whether to be a signatory or not. Non-signatories do not contribute (or contribute less), while signatories adopt the strategy that maximizes their welfare (minimizes their expected cost) as a whole. An agreement is self-enforcing if no signatory has an incentive to opt-out and become a non-signatory, and no

non-signatory has an incentive to opt-in and become a signatory[21]. Note that if the optimal solution is a Nash equilibrium, it is also a SEA. However, there could be cases in which the optimal solution is a SEA but not a Nash equilibrium. For example, in the context of our game, consider the case in which $s_i = 2$ by all countries is not a Nash equilibrium: if all countries adopt $s_i = 2$, it may benefit country $j$ to deviate to, say, $s_j = 1$. However, following this opting out by country $j$, the best strategy of the remaining countries (the signatories) may become $s_i = 1$ instead of $s_i = 2$. If country $j$ anticipates that the remaining countries will reduce their contribution to $s_i = 1$ in response to its own deviation, it might be beneficial for country $j$ to persist with $s_i = 2$.

**Numerical methods**. To calculate the optimal solution, we calculated the expected cost to a given country $i$, $C_i$, given the set of strategies that all the countries adopt, we first calculate $P_1$ and $P_2$ (Eq. (1)), and then we set $P_{\geq 3} = 1 - P_1 - P_2$ and substitute the results in Eq. (2). In turn, we examine all possible sets of strategies, where in each set, each country can contribute a different amount ($s_i$ equals 0, 1, or 2 for each $i$). For each set, we calculate the total cost (Eq. (3)), and we find the set for which the total cost is minimized.

Next, we did the following to check whether the optimal solution is a Nash equilibrium. Without loss of generality, we examine whether country 1 can benefit from deviating if all the countries adopt the optimal solution, $s_i = s^{opt}$. Specifically, country 1 has 2 possible deviations: to $s_1 = 0$ or $s_1 = 1$ if $s^{opt} = 2$, and to $s_1 = 0$ or $s_1 = 2$ if $s^{opt} = 1$. We first calculate $C_1^{opt}$, the expected cost to country 1 if it does not deviate (and the other countries adopt $s_i = s^{opt}$). Then, we calculate $C_1^{dev1}$ and $C_1^{dev2}$, the expected cost to country 1 following each of its possible deviations. We conclude that the optimal solution is a Nash equilibrium if and only if $C_1^{opt} \leq C_1^{dev1}$ and $C_1^{opt} \leq C_1^{dev2}$.

In turn, we did the following to check whether the optimal solution is a self-enforcing international agreement. For every $1 \leq n \leq N$, we consider $n$ signatories and $N - n$ non-signatories, and we calculate the utility (minus the cost – Eq. (2)) of a signatory, $u_s(n)$, and the utility of a non-signatory, $u_o(n)$. (Note that, for each $n$, $u_o(n) \geq u_s(n)$[21]). In turn, an agreement with $n = N$ signatories is by definition the optimal solution. This solution is a self-enforcing international agreement if a country cannot benefit from opting-out when all $N$ countries are signatories. Specifically, if a country remains a signatory, its utility is $u_s(N)$, while if it opts-out, its utility becomes $u_o(N - 1)$. Therefore, the optimal solution is self-enforcing if and only if $u_s(N) \geq u_o(N - 1)$.

**Statistics and reproducibility**. This study is quantitative mathematical and computational. No statistical method is used. All results are fully reproducible: The methods are fully described in the Methods section and parameter values are given in the caption of each relevant figure.

**Reporting summary**. Further information on research design is available in the Nature Research Reporting Summary linked to this article.

## Results

### Optimal solution varies between contribution, no contribution, and partial contribution, depending on parameters. We first examine the optimal solution from the point of view of a social planner who aims to maximize the social welfare of all vaccine-rich countries combined. Specifically, we find the contribution that maximizes social welfare (minimizes the expected cost) of the $N$ vaccine-rich countries as a whole across the various

parameter ranges. Due to the substantial uncertainty regarding some parameter values, such as the expected rate of VOC ($\lambda$), the analysis considers the following parameter ranges: $2 \leq N \leq 10$, $0 \leq \alpha \leq 0.5$, $0 \leq v_{max} \leq 1$, and $0 < \lambda \leq 2$. Parameter ranges were chosen based on estimates of their plausible values in the context of the COVID-19 pandemic (see: Supplementary Note 1). However, note that the model is not limited to these specific parameter ranges. The top panel in Fig. 2 maps the optimal solutions and their stability to parameters $v_{max}$ (Y-axis) and $\lambda$ (X-axis), holding the number of countries and effectiveness of stocking constant ($N = 6$ and $\alpha = 0.4$, respectively). In turn, each panel in Figs. 3 and 4 also shows the optimal solutions and their stability as a function of $\alpha$ and $N$.

Our results show that the optimal solution is always an equal donation by all countries, of either 0, 1, or 2 full-population rounds of vaccinations, depending on the values of the parameters $\lambda$, $\alpha$, $N$, and $v_{max}$ (Figs. 2–4). Specifically, zero donation of surplus vaccines by all countries ($s_i = 0$ for all $i$) is optimal in parameter regions where $\lambda$ is sufficiently large and $v_{max}$ and $\alpha$ are sufficiently small (black region $VII$ in Figs. 2–4). Full donation ($s_i = 2$ for all $i$) is optimal if $v_{max}$ is sufficiently close to 1 and/or $\alpha$ is sufficiently large (yellow and yellow-grey regions $I$, $II$, and $III$ in Figs. 2–4). In turn, donation of half of the surplus vaccines ($s_i = 1$ for all $i$) is optimal if $\lambda$ is sufficiently small, and $v_{max}$ and $\alpha$ have some intermediate values (blue and blue-grey regions $IV$, $V$, and $VI$ in Figs. 2–4). Note that it is not trivial that the optimum solutions do not include one in which different countries choose different strategies. This result follows as there is positive feedback between the marginal benefit from donation and the amount donated by other countries; therefore, if it is beneficial for the countries to increase their donation from $s_i = 0$ to $s_i = 1$, then it is even more beneficial to increase the donation if some countries already donate.

Not surprisingly, vaccine-rich countries benefit more from donating if $v_{max}$ is large. If $v_{max}$ is much below 1, then even if all the extra vaccine doses are donated, only a small portion of the world's unvaccinated population can be covered, and the risk of future variants cannot be significantly reduced. On the other hand, if $v_{max}$ is close to 1, nearly the entire unvaccinated world population can get vaccinated, thus donating countries can expect to reduce the risk of variants considerably. Interestingly, however, our results indicate that the threshold of $v_{max}$ where the optimal strategy becomes $s_i = 2$, is not very sensitive to $\lambda$ – as shown, for example, in Fig. 2, where $v_{max} > 0.75$.

Below this threshold of $v_{max}$, $\lambda$ acquires a substantial effect on whether $s_i = 1$ or $s_i = 0$ is optimal. This is relevant from a policy perspective as it means that in quite realistic ranges of $v_{max}$, whether it is beneficial for the vaccine-rich countries to donate an extra dose per capita or not donate at all depends to a large extent on the expected frequency of VOCs. Specifically, if $\lambda$ is small, it may be beneficial for vaccine-rich countries to contribute one dose per capita each, even under relatively small values of $\alpha$ and $v_{max}$ – as depicted by the blue regions ($IV$, $V$, and $VI$) in Fig. 2. By contrast, if $\lambda$ is large, donation is beneficial only if the value of $v_{max}$ is larger.

These patterns are modified, yet retain their basic structure, when the unavoidable fraction of the cost of a future outbreak ($\alpha$) varies. Vaccine-rich countries are clearly better off internationally donating more surplus vaccines when $\alpha$ rises. In the extreme case in which the time it takes to administer pre-stocked extra doses is long enough to incur the total cost of a new outbreak ($\alpha = 1$), then stocking extra doses domestically is practically worthless, and a better strategy is to donate all surplus doses to the vaccine-poor world to reduce the risk of variants. However, as $\alpha$ decreases, it becomes more worthwhile to hold enough doses for an extra domestic vaccination round. Figure 3 illustrates this with $\alpha$ values

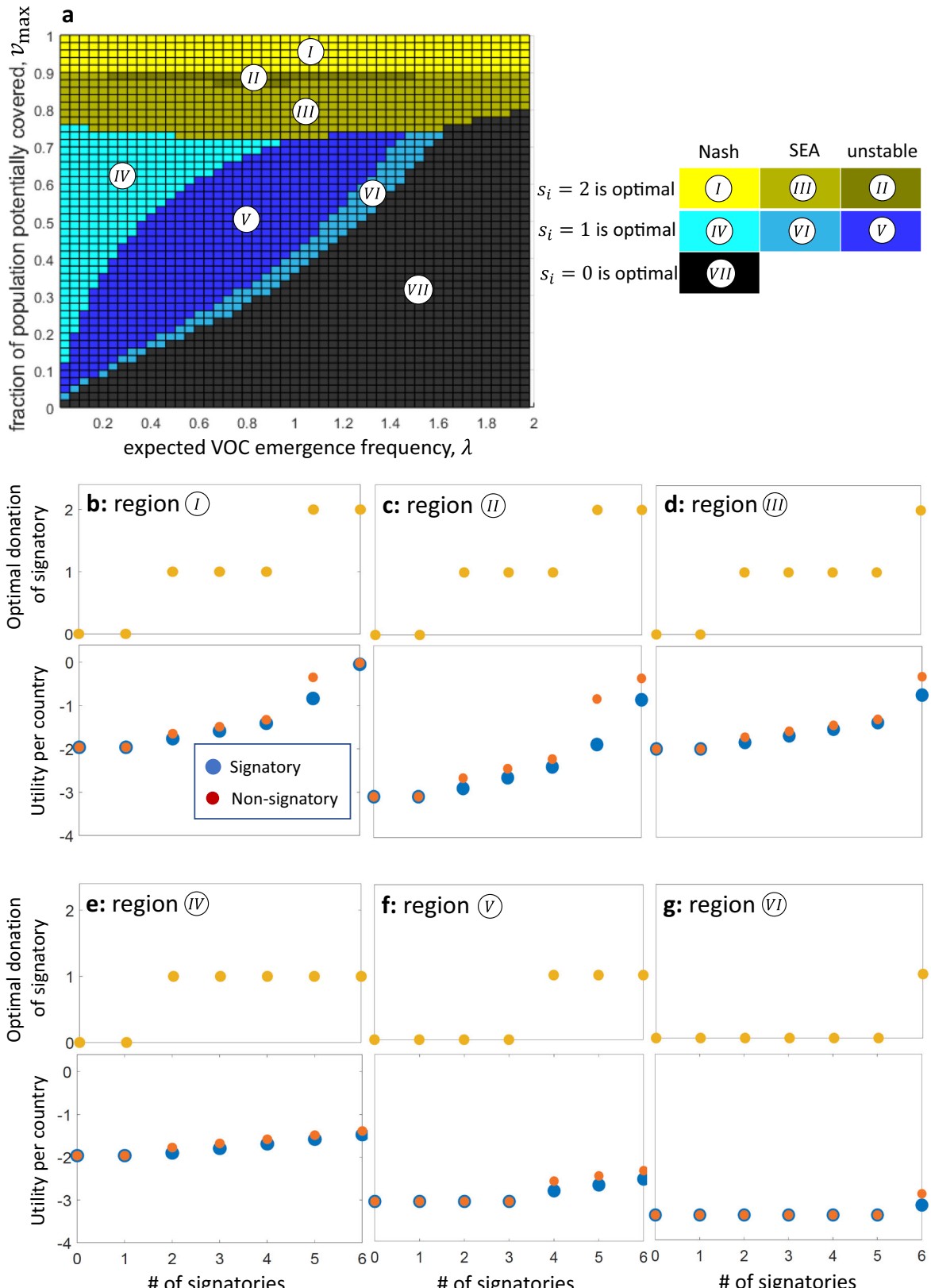

of 0.1, 0.2, and 0.3. Not surprisingly, the surfaces of the two-dose-optimal regions (I, II, and III) and the one-dose-optimal regions (IV, V, and VI) are smaller for lower $\alpha$ values. Also note that it follows directly from the model that the optimal solution does not depend on the number of vaccine-rich countries, $N$, as long as $v_{max}$ is held constant, as shown in Fig. 4. The social planner aims

to minimize the cost to the vaccine-rich world's population as a whole, as if it were a single country. In turn, from the social planner's perspective, $N$ is only a constant that multiples the utility function, which cannot affect the optimal solution[22]. Therefore, the optimal solution does not depend on the number of countries that make up the vaccine-rich world (Fig. 4).

**Fig. 2 The optimal solutions and their stability across parameters.** The optimal solution and its stability depend on the values of the parameters $v_{max}$ and $\lambda$. The optimal solution is given by the strategies $s_i$ that minimize the total expected cost to all vaccine-rich countries as a whole (Eq. (3)). **a** Each colored rectangle on the top panel represents the solution for the corresponding values of $\lambda$ and $v_{max}$, where the other parameters are $\alpha = 0.4$ and $N = 6$. For some parameter regions, denoted as regions *I*, *II*, and *III* (yellow shades), $s_i = 2$ by all vaccine-rich countries is optimal. For some other parameter regions, denoted as regions *IV*, *V*, and *VI* (blue shades), $s_i = 1$ by all countries is optimal. Otherwise (region *VII*, black), $s_i = 0$ by all countries is optimal. The parameters also determine whether the optimal solution is stable and can be adopted by rational agents, where each country acts self-interestedly. In regions *I* and *IV*, the optimal solution is also a Nash equilibrium. In turn, in regions *III* and *VI*, the solution is not a Nash equilibrium but is a self-enforcing international agreement (SEA). (Note that if the optimal solution is a Nash equilibrium, it is also an SEA.) **b–g** The optimal solution (yellow dots), the utility of signatories (blue dots) and the utility for non-signatories (red dots) are shown for each region as the function of the number of signatories ($n$). This can determine whether the optimal solution is also an SEA: In regions *I*, *III*, *IV*, and *VI*, the utility of signatories when $n = N = 6$ is higher than the utility of a non-signatory when $n = 5$; therefore, it does not worth for a signatory to opt-out, and $n = N$ (the optimal solution) is also an SEA.

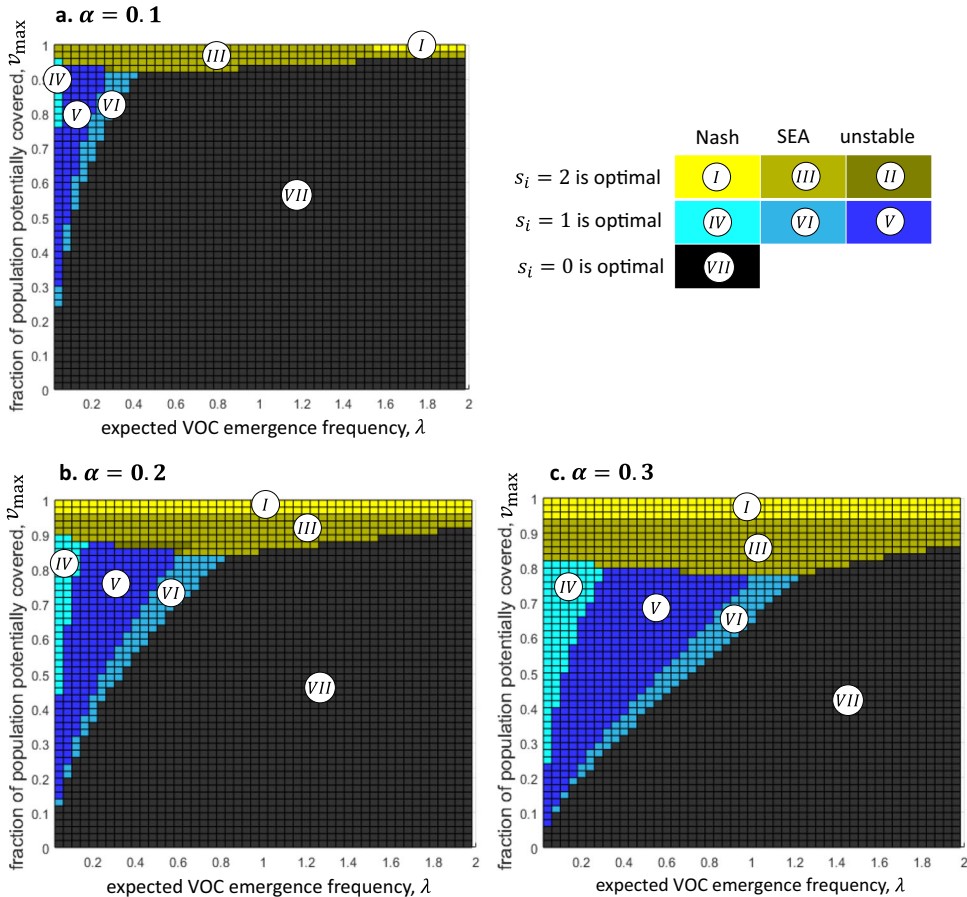

**Fig. 3 The optimal solution and its stability across $\alpha$ values.** The optimal solution and its stability depend on the values of the parameters $v_{max}$, $\lambda$, and $\alpha$. Each panel characterizes a different value of $\alpha$: **a** $\alpha = 0.1$. **b** $\alpha = 0.2$. **c** $\alpha = 0.3$. $N = 6$ in all panels. Each colored rectangle on the top panel represents the solution for the corresponding values of $\lambda$ and $v_{max}$. In regions *I*, *II*, and *III* (yellow shades), $s_i = 2$ by all vaccine-rich countries is optimal. In regions *IV*, *V*, and *VI* (blue shades), $s_i = 1$ by all countries is optimal. In region *VII* (black), $s_i = 0$ by all countries is optimal. One notable observation is that if $\alpha$ (the unavoidable fraction of the cost of an outbreak, given surplus doses) is smaller, then it is less worthwhile to donate, and the range of parameters that renders donations advantageous is narrower.

**Some optimal solutions are Nash equilibria or self-enforcing international agreements**. Having identified optimal strategies, we next examine whether one could expect the $N$ vaccine-rich countries to adopt these optimal strategies, acting strictly self-interestedly and voluntarily. Again, the answer depends on the parameter values for $\lambda$, $\alpha$, $v_{max}$, and $N$ (Figs. 2–4).

*Full donation.* In some parameter regions (denoted as region *I* in Figs. 2–4), donating both surplus vaccine doses is not only optimal but is also a Nash equilibrium: if $N - 1$ countries adopt $s_i = 2$, then the best response of the last country is also $s_i = 2$. In that case, we expect the $N$ countries to willingly contribute both

their surplus full-population stocks to vaccine-poor countries. In other words, under configurations of high values of $v_{max}$ and realistic values of $\alpha$ (see Supplementary Note 1 and Supplementary Table 1), double-dose donation by vaccine-rich countries is a Nash equilibrium.

Furthermore, in certain parameter regions in which donating $s_i = 2$ doses is optimal but is not a Nash equilibrium, there still remain realistic prospects for stable donation strategies by vaccine-rich countries, as the optimal solution may be a self-enforcing international agreement (SEA). The $N^{th}$ country might not be willing to deviate from an agreement to donate $s_i = 2$ if it foresees that the optimal contribution by the remaining $N - 1$

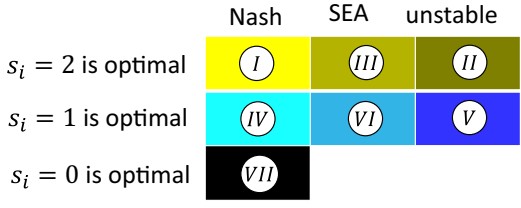

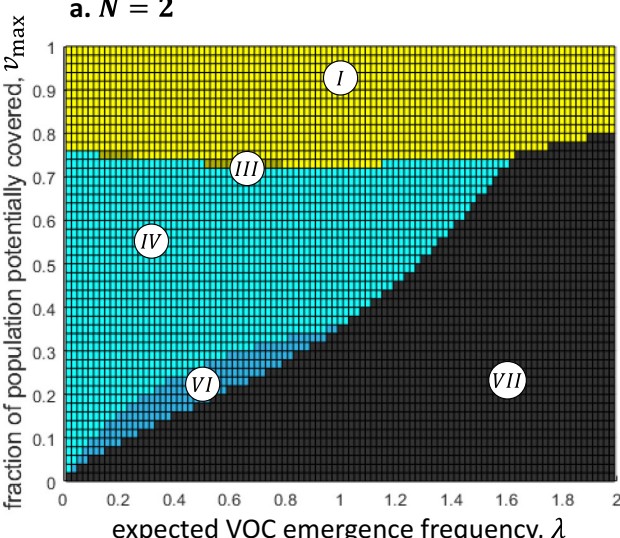

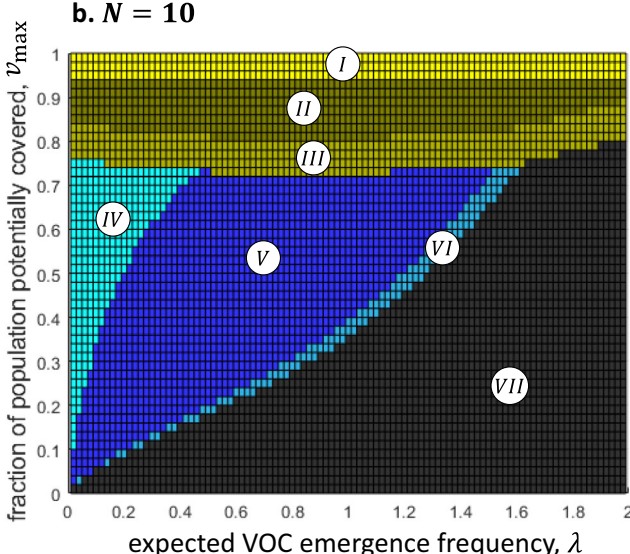

**Fig. 4 The stability of the optimal solution depends on the number of vaccine-rich countries.** The optimal solution does not depend on the number of vaccine-rich countries, $N$, as long as $v_{max}$ is held constant; however, the optimal solution is stable for a wider range of parameters if $N$ is smaller. Each colored rectangle on the top panel represents the solution for the corresponding values of $\lambda$ and $v_{max}$. In regions $I$, $II$, and $III$ (yellow shades), $s_i = 2$ by all vaccine-rich countries is optimal. In regions $IV$, $V$, and $VI$ (blue shades), $s_i = 1$ by all countries is optimal. In region $VII$ (black), $s_i = 0$ by all countries is optimal. The only difference in parameters between the panels is that $N = 2$ in panel (**a**) and $N = 10$ in panel (**b**). $\alpha = 0.4$ in both panels. The case of $N = 6$ is demonstrated in Fig. 2. Note that in panel (**a**), where $N = 2$, the optimal solution is stable (Nash equilibrium or self-enforcing international agreement (SEA)) for all values of $v_{max}$ and $\lambda$; but in panel (**b**), where $N = 10$, there are parameter regions in which the optimal solution is unstable (regions $II$ and $V$).

countries will become $s_i = 1$ (region $III$ in Fig. 2). This idea is demonstrated in the bottom panel of Fig. 2. If the 6th country opts out, the other 5 countries will now be better off adopting a strategy in which $s_i$ equals 1 rather than 2 (yellow dot for 5 signatories is lower than for 6 signatories), rendering the utility for the putative non-signatory (6th country) lower than its utility had it not opted-out of the agreement (the red dot with 5 signatories is lower than the blue dot with 6 signatories). In other words, in addition to being a Nash equilibrium, under a still larger set of configurations of realistic values of both $\alpha$ and $v_{max}$, a double-dose donation can also be a SEA.

*Partial donation.* Similarly, for some parameter values (the light blue regions $IV$ and $VI$ in Figs. 2–4), donating one dose is both optimal and stable. Specifically, $s_i = 1$ by all countries is a Nash equilibrium in region $IV$. In addition, if $s_i = 1$ is optimal but not a Nash equilibrium, $s_i = 1$ by all countries is self-enforcing in parameter region $VI$. Note that SEAs that are not Nash equilibria appear in parameter values that are close to those where the optimal strategy becomes $s_i = 1$ instead of $s_i = 2$ (region $III$), or becomes $s_i = 0$ instead of $s_i = 1$ (region $VI$). The reason is that, in those regions, the deviation of one country can change the optimal strategy of the remaining $N - 1$ countries (see lower panels in Fig. 2).

As Fig. 4 shows, the optimal solution is stable for a wider range of parameters if the number of vaccine-rich countries, $N$, is smaller. Increasing $N$ from 2 to 10, while holding the other parameter values constant, diminishes the set of parameter configurations in which both $s_i = 2$ and $s_i = 1$ are a Nash equilibrium (regions $I$ and $IV$, respectively), as depicted in Fig. 4. The reduction in the range of parameters in which donations are a stable strategy as $N$ increases occurs because the benefit to a given country due to its own contribution becomes smaller, while the cost per capita to the donating country remains the same.

Finally, note that in other regions, donation of two doses (region $II$) or one dose (region $V$) is optimal (maximizes the social welfare of the vaccine-rich countries as a whole), but the optimal solution is unstable. Instead, the stable solutions in these parameter regions are sub-optimal and dictate donation by only some countries. Figure 5 shows the number of countries that do not donate following the SEA solution (Fig. 5a), and the corresponding "social efficiency deficit" or "cost of anarchy"–shows the difference between the cost of following the optimal solution and the cost of following the SEA solution[23–25] (Fig. 5b).

## Discussion

This study adopts a game-theoretical approach to explore the conditions under which strictly self-interested vaccine-rich countries would voluntarily donate their surplus vaccine doses to vaccine-poor countries in a pandemic context, without recourse to any other motivations such as international solidarity. We addressed this question by first identifying the optimal vaccine donation strategies for vaccine-rich countries and then examining whether these optimal strategies could be adopted by these vaccine-rich countries, assuming that they act as strictly self-interested, rational agents.

We show that full donations are optimal when their potential impact is high ($v_{max}$ is large), and when stocking surplus doses is less effective in averting the cost of a future outbreak ($\alpha$ is large). Specifically, if the fraction of the unvaccinated world population that can be vaccinated if all vaccine-rich countries fully donate ($v_{max}$) is above a certain threshold, full donation is optimal. The value of this threshold of $v_{max}$ is determined by $\alpha$. Interestingly, above this threshold, the optimal solution of full donations is not sensitive to the expected annual rate of VOCs ($\lambda$). Below this

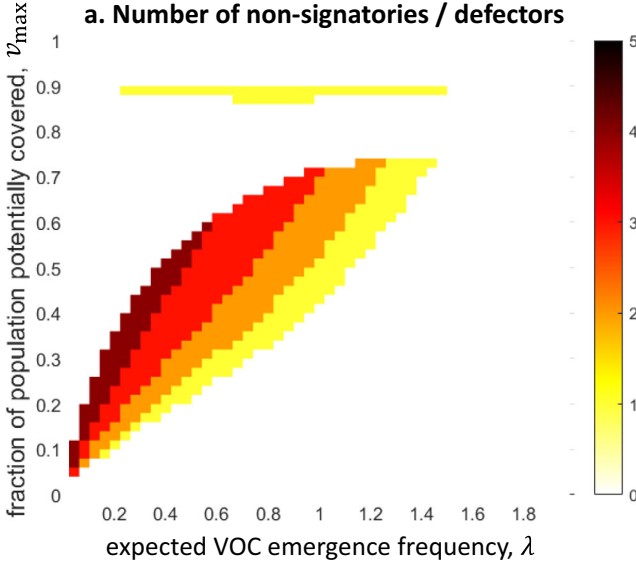

**a. Number of non-signatories / defectors**

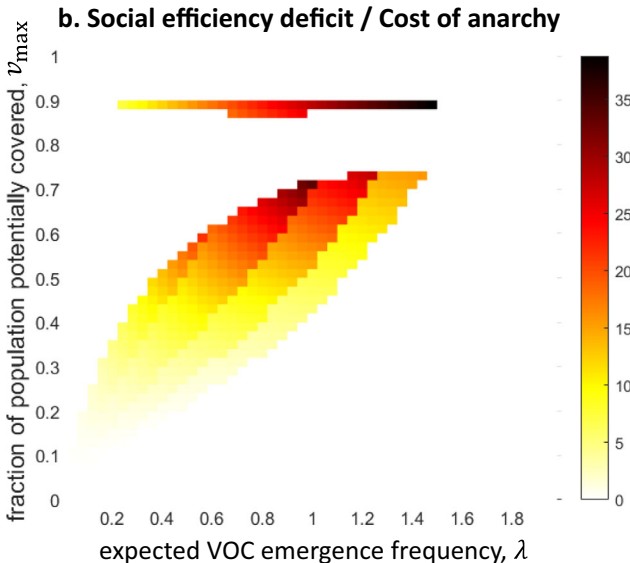

**b. Social efficiency deficit / Cost of anarchy**

**Fig. 5 If the optimal solution is unstable, the stable solution exhibits a social efficiency deficit.** Demonstrated are the differences between the optimal solution and the stable, self-enforcing international agreement (SEA) solution. **a** The number of countries that opt-out of the international agreement (non-signatories) is zero if the optimal solution is also a stable SEA solution (white region). Otherwise, however, the SEA solution comprises one or more non-signatories (red color scale). **b** The "social efficiency deficit" or the "cost of anarchy" is defined as the difference between the total cost following the optimal solution and the total cost following the stable, SEA solution (red color scale). The social efficiency deficit is zero for those parameter values where the optimal solution is also an SEA, but it is positive otherwise. Note that we used the same parameters as in Fig. 2, and the parameter regions where the maps are colored is the same as regions *II* and *V* in Fig. 2, which are the regions where the optimal solution is unstable.

threshold, partial donations or no donations are optimal, depending jointly on the rate of VOCs ($\lambda$), $v_{max}$, and the unavoidable fraction ($\alpha$) of the cost of future outbreaks (despite stocking surplus doses). These results point to the importance of $v_{max}$. Although the expected number of variants ex-ante affects the utility of donations, it is also ex-post influenced by donations, depending on their potential impact on the unvaccinated world

population. If $v_{max}$ is sufficiently large, the pre-donation rate of expected variants becomes inconsequential for the utility of donations, as reflected by the results. However, if $v_{max}$ is relatively low, pre-donation rate of expected variants is an important determinant of the utility of vaccine donation.

This study's second and crucial finding is that in certain parameter regions, self-interested vaccine-rich countries can be expected to willingly donate the socially optimal number of surplus vaccine doses. That is, we show that these optimal surplus donation strategies are also stable. Specifically, when *full* (double-dose) donation is optimal, it is also a Nash equilibrium if $v_{max}$ is sufficiently large (region *I* in Figs. 2–4), and it is a self-enforcing agreement in a somewhat lower range of $v_{max}$ (region *III*). When *partial* (single-dose) donation is optimal, it is also a Nash equilibrium if the risk of future variants of concern ($\lambda$) is relatively low and $v_{max}$ is at medium levels (region *IV*), and it is a self-enforcing agreement in a narrower set of parameter regions (region *VI*). Finally, while the number of potential donor countries ($N$) does not affect the optimal solution (assuming that $v_{max}$ is constant), a larger number of countries implies a narrower range of parameter values in which the optimal solution is stable (Fig. 4). Thus, if the aggregate stock of surplus doses of vaccine-rich countries is distributed over a large number of such countries, donations are less likely to be a stable strategy even when they are optimal.

Tentative policy implications can be inferred from these findings. Our results offer hope for global effective pandemic solutions because they identify significant windows in which significant self-interested vaccine donation is viable and stable. That said, we also see large parameter ranges where the prediction is zero donations. Given our assumption that donations are feasible only when they are the optimal solution, within the parameter ranges where this is the case, coordination efforts are likely to be fruitful. The results of this study may contribute to the efforts of organizations such as the World Health Organization in promoting such endeavors.

This model is intended to address the problem of country-level vaccine donations in the context of a pandemic in general. Applying it to a particular case, such as the COVID-19 pandemic, requires more accurate estimates of the relevant parameters than are currently available. Still, drawing on rough estimates of our model parameters in this context (see Supplementary Note 1) predicts a stable partial donation by the US and the EU. Existing data suggests that EU countries and the US have donated about 900 million vaccine doses (https://launchandscalefaster.org/covid-19/vaccinedonations. Data extracted on April 5, 2022). This represents a sizable donation, although it is a somewhat lower estimate than our model predicts. Given that existing donation data is likely underestimating the true amounts, and the fact that donations are still ongoing, these data offer preliminary and tentative credence to the model predictions.

The vaccine donation game presented here indicates a greater potential for international cooperation, compared, for example, to climate agreements, due to the positive feedback between the marginal benefit from donation and the amount donated by other countries. Specifically, if a given country donates, it decreases the number of unvaccinated people worldwide, which makes the contribution of any other country more valuable because its donation now covers a greater portion of the unvaccinated world's population. Therefore, the contribution of a given country increases other countries' incentive to contribute. This is fundamentally different from the case of international climate agreements[21,26], in which the contribution of some countries makes it less beneficial for more countries to join a treaty because climate cost is assumed to increase super-linearly with temperature changes.

The literature on vaccine donation games goes back to[13], who examined the international subsidy of Smallpox vaccine in low-income countries. That problem was, however, fundamentally different from ours and characterized a much later stage of the pandemic. There was no question about the optimality of the subsidy, and there was no shortage of vaccine doses. Instead, the question was which country would be the one to invest the money[13]. Furthermore, a study of international subsidy for the treatment of harmful invasive species in low-income countries to prevent their spread into high-income countries, showed that, in Nash equilibrium, fewer countries might do a better job than multiple countries[27]. In that study, however, a single or a few high-income countries had the capacity to eliminate the species even when working alone. Finally, in the context of the COVID-19 pandemic, a study of the influence of countries' policies on each other, focusing on pre-vaccine COVID-19 preventative measures, concluded that the game is analogous to a weakest-link game: a few countries that do not take sufficient preventative measures are sufficient to spread the pandemic internationally[28]. Note that the vaccine donation game developed here is fundamentally different and does not necessitate contribution from all countries.

Our analysis is intended to offer a general game-theoretical approach to solving the problem of donating surplus vaccines during a pandemic. It unavoidably incorporates several simplifying assumptions. First, we assume that countries stock excess vaccine doses for the purpose of responding quickly to an outbreak that is due to a VOC. However, countries also stock vaccines due to uncertainty regarding the duration of vaccine efficacy, or in order to diversify the stock of vaccines (by producers and technologies) due to uncertainty regarding their consequent approval and efficacy. These additional reasons for stocking surplus vaccine doses are not related to the risk of VOC, and thus to the extent that they are dominant, may decrease the likelihood of donations.

Stocking vaccine doses in practice may be implemented in several ways. It can be done through the actual stocking of vaccine vials within the country's territory, but this method is often limited due to vaccine storage periods. A second method is to obtain purchasing contracts that prioritize the supply of additional vaccines, if required. Donations are possible in all three methods. However, the different methods entail varying response times during an outbreak, with the potentially quickest response in the case of physical stocks, and the longest in the case of a dedicated budget. These different response durations are captured in our model by $\alpha$; thus, specific information about the stocking methods of each vaccine-rich country can be used to enhance the estimates in concrete circumstances.

Lastly, vaccine-rich countries in our model are assumed to be equal in size. Relaxing this assumption may change some of the results. Notably, it is likely that the optimal strategy will not dictate the same for the entire set of vaccine-rich countries. Moreover, variation in vaccine-rich country sizes may alter the results regarding solution stability. Further research is required in order to apply the model to real-life differences between vaccine-rich countries.

## Data availability
There are no restrictions on the data used in our paper. The source data for Fig. 1 is given as Supplementary Data 1 and can also be found in Our World in Data https://github.com/owid/covid-19-data/tree/master/public/data. Figures 2–5 show the results in their entirety, and source data can also be found in Supplementary Data 2.

## Code availability
The description of the algorithm is given in detail in Methods: numerical methods. This section has sufficient information to reproduce our results in full. In addition, our code is available online[29] (https://zenodo.org/record/6912601#.YuvooXZBxD8).

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

## Acknowledgements
We thank the seminar participants at the Hebrew University Federmann School of Public Policy, the editor of Communications Medicine, and the anonymous reviewers for their

insightful comments. We acknowledge the generous support of the Volkswagen Stiftung, grant no. Az. 99 057.

## Author contributions

A.L., R.S.K., P.V., and M.T. conceptualized the study and developed the model; A.L. developed the code and performed the analysis; A.L., R.S.K., P.V., and M.T. wrote the paper.

## Competing interests

The authors declare no competing interests.
