## [Peer Review File · Communications Medicine]

Reviewers' comments:

Reviewer #2 (Remarks to the Author):

This work concerns on a specific situation around 'vaccine-diplomacy' from the game theoretical standpoint, which seems new, no one has ever explored. In this sense, I can positively evaluate this work.

A vaccine-rich country takes strategy $s=\{0,1,2\}$ implying the amount of surplus doses she stored to donate vaccine-poor countries. This is not simply donation; just wasting her money, because if poor countries fail to supply vaccine, the probability of a variant strain emerging increases, which also impacts (may devastates) rich countries. Thus, as a whole, the situation encompasses a social dilemma, which might be same as the so-called public-goods game-type dilemma, i.e., either Defector ($s=0$ is D-strategy) – dominate (like Prisoner's Dilemma) or Polymorphic (co-existence of C and D; like Chicken game).

Such a story seems somehow attracting to the audience.

The result they explored visually suggests optimal state along the parameter of variant emerging (λ) and coverage of vaccination (v_{max}).

I'm bit confused whether why the authors did not simply show Nash Equilibrium (NE) besides those social optimal (SO) state. If there is a gap between NE and SO visually, it is literally working as a hard evidence of existence of social dilemma. Some previous study gives theoretical framework of this gap to measure 'social dilemma', called Social Efficiency Deficit (SED). I strongly suggest the authors to extend their visual result to such discussion by referring to the concept of SED and relevant literatures (for example; (i) Social efficiency deficit deciphers social dilemmas, Scientific Reports 10, 16092, 2020).

When it comes to the term 'vaccination game' as one of the jargons is the merged framework of mathematical epidemiology, quantifying disease spreading, with evolutionary game theory, where the dynamics of each agent's decision, e.g., whether he/ she committing a vaccination or not, is fairly quantified. Such previous studies are different from what the authors contributing in the present study. And the authors should introduce those by citing several precursors. There are several review papers and books; e.g. (ii) Sociophysics Approach to Epidemics, Springer, 2021.

This paper introduces a game-theoretic approach to explore the conditions under which strictly self-interested vaccine-rich countries would voluntarily donate their surplus vaccine doses to vaccine-poor countries in the context of a pandemic. This research work is quite interesting and nice presented. I have some suggestions to further improve its quality.

1) The author assumes that 1 extra dose may protect against 1 variant and 2 doses may protect against 2 variants. But the effectiveness of the same vaccine against different mutant strains is different. Therefore, should different vaccine effectiveness be considered?

2) The fig. 4 shows that the optimal solution does not depend on the number of rich countries. This conclusion seems to be obtained only from the figures. The author should make a relevant theoretical proof for it.

3) The proposed model is from the point of view of a 'rich-world' social planner who aims to maximize the social welfare of all vaccine-rich countries combined. If we start from the overall situation and consider the interests of all countries, will the model and conclusion be different from the previous ones? The author should explain it.

4) In Figure 4(a), why are there no area 2 and area 5? The author should explain it.

5) The value of s_i is 0, 1 or 2. Then whether s_i can take a decimal value when some vaccine-rich countries choose to donate extra dose and some countries choose not to donate.

6) If $v < 1$, the actual number of VOCs that emerge is given by a Poisson distribution with a mean $\lambda(1-v)$. Please explain why a Poisson distribution is chosen to characterize the probability for the occurrence of exactly k variants.

7) Although the conclusion drawn from the numerical studies are given, what is the insights for the decision maker? At present, are there any relevant cases to support this paper?

Reviewer 1

We sincerely thank the Reviewer for carefully reading our manuscript and for the helpful comments. Below please find the Reviewer's comment (black fonts) followed by our response (blue & bold fonts).

This paper introduces a game-theoretic approach to explore the conditions under which strictly self-interested vaccine-rich countries would voluntarily donate their surplus vaccine doses to vaccine-poor countries in the context of a pandemic. This research work is quite interesting and nicely presented. I have some suggestions to further improve its quality.

We thank the Reviewer for finding our work interesting and well-presented.

1) The author assumes that 1 extra dose may protect against 1 variant and 2 doses may protect against 2 variants. But the effectiveness of the same vaccine against different mutant strains is different. Therefore, should different vaccine effectiveness be considered?

The Reviewer correctly comments that we assume that the ineffectiveness of the vaccine, α , is the same against the first two variants (if more than one variant emerges). Clearly, this is a simplifying assumption, but a reasonable one. Due to the considerable uncertainty about the magnitude of α , we consider the implications of a range of values for α of 0.1 to 0.5. In the particular context of the COVID-19 pandemic, we consider a narrower plausible range of 0.25-0.5 (see pp. 17-18 and supplementary information, pages 1-2).

Our analysis shows that the results when considering α_1 for the first outbreak and α_2 for the second outbreak are very similar to those in our model that includes a single α with an intermediate value for both outbreaks.

In the revised Manuscript, we added a clarification stating that considering a single value of α is a simplifying assumption: "For simplicity, we assume that if there is more than one outbreak, the cost alpha is the same in the first and the second outbreaks" (footnote 7, page 8).

2) The fig. 4 shows that the optimal solution does not depend on the number of rich countries. This conclusion seems to be obtained only from the figures. The author should make a relevant theoretical proof for it.

The Reviewer correctly points that, in Fig. 5 (Fig. 4 in the original submission), the socially optimal solution does not depend on the number of countries, N . The reason is that, in Fig. 5, the choice of v_{\max} is independent of the choice of N : Namely, we assume that each country, if adopting the strategy $s_i = 2$, contributes an amount that suffices for v_{\max}/N of the unvaccinated world's population. In turn, from the standpoint of a social planner that aims to minimize the total cost to all vaccine-rich countries, the optimal solution depends on v_{\max} . If v_{\max} is held constant, the optimal solution doesn't matter how many countries there are.

In the revised Manuscript, we clarify this point: “... it follows directly from the model that the optimal solution does not depend on the number of rich countries, N , as long as v_{\max} is held constant, as shown in Fig. 5: The social planner aims to minimize the cost to the vaccine-rich world’s population as a whole, as if it were a single country. In turn, from the social planner’s perspective, N is only a constant that multiplies the utility function, which cannot affect the optimal solution (Tadelis 2013). Therefore, the optimal solution does not depend on the number of countries that the rich world (Fig. 5).” (see: bottom of page 12).

3) The proposed model is from the point of view of a ‘rich-world’ social planner who aims to maximize the social welfare of all vaccine-rich countries combined. If we start from the overall situation and consider the interests of all countries, will the model and conclusion be different from the previous ones? The author should explain it.

If we understand this question correctly, the Reviewer inquires whether there is a difference between (1) a “rich-world social planner,” who aims to maximize the welfare of all vaccine-rich countries, and (2) a “whole-world social planner,” who aims to maximize the welfare of all the countries, both rich ones that donate vaccines and poor ones that only receive the donation.

The answer is that, indeed, there is a fundamental difference between the two. In our study, we considered only the “rich-world social planner.” To clarify this in the text, we explain that “given that vaccine donations are dependent on the choices of vaccine-rich countries, the analysis strictly takes these countries’ perspective.” (p. 5). We further note that some previous studies have addressed the question of optimal vaccine allocation from a global perspective (footnote 4).

4) In Figure 4(a), why are there no area 2 and area 5? The author should explain it.

Fig. 5a (Fig. 4a in the original submission) shows a case where the number of vaccine-rich countries is very small ($N = 2$). In turn, as we explain in the text, fewer countries imply that it is more likely that the optimal solution is also stable, as shown in Fig. 5. (The case $N = 1$ is the case where the optimal solution is the only stable solution by definition, but as N increases, the self-interests of countries may play a greater role.)

In the caption of Fig. 5, we explain that “the optimal solution is stable for a wider range of parameters if N is smaller.” We also added: “Note that in panel A, where $N = 2$, the optimal solution is stable (Nash equilibrium or SEA) for all values of v_{\max} and λ ; but in panel B, where $N = 10$, there are parameter regions in which the optimal solution is unstable (regions 2 and 5)”.

5) The value of s_i is 0, 1 or 2. Then whether s_i can take a decimal value when some vaccine-rich countries choose to donate extra dose and some countries choose not to donate.

The strategy s_i denotes the decision of country i and can be 0, 1, or 2. However, it is possible that some countries donate one dose and other countries donate two doses. For example,

the case where country 1 donates one dose and country 2 donates two doses is denoted by $s_1 = 1$ and $s_2 = 2$. Note, however, that our results show that the optimal solution always comprises the same contribution by all the vaccine-rich countries. At the same time, if the stable solution, SEA, differs from the optimum, it is indeed possible that some countries “free ride” and contribute less than other countries. This is demonstrated in a new figure, Fig. 3A, which demonstrates the number of countries that do not contribute in a SEA.

In the Manuscript, when we explain how we find the optimal solution, we explain that “we examine all possible sets of strategies, where in each set, each country can contribute a different amount (s_i equals 0, 1, or 2 for each i). For each set, we calculate the total cost (Eq. (3)), and we find the set for which the total cost is minimized” (Methods section, “calculating the optimal solution” subsection, page 29).

Also, we explain in footnote 9 (page 11) that “it is not trivial that the optimum solutions do not include one in which different countries choose different strategies. This result follows as there is a positive feedback between the marginal benefit from donation and the amount donated by other countries; therefore, if it is beneficial for the countries to increase their donation from $s_i = 0$ to $s_i = 1$, then it is even more beneficial to increase the donation if some countries already donate.”

6) If $v < 1$, the actual number of VOCs that emerge is given by a Poisson distribution with a mean $\lambda(1 - v)$. Please explain why a Poisson distribution is chosen to characterize the probability for the occurrence of exactly k variants.

We assume that variants are independent of one another: The probability of a variant to emerge does not depend on the number of other variants that emerge (equivalently, the emergence of a variant does not change the probability that other variants emerge). Therefore, by definition, the probability distribution of the number of variants is given by the Poisson distribution (Loeve 2017). This is clarified in the revised manuscript in the bottom paragraph on page 7.

7) Although the conclusion drawn from the numerical studies are given, what is the insights for the decision maker? At present, are there any relevant cases to support this paper?

This model is intended to address the problem of country-level vaccine donations in the context of a pandemic in general. Furthermore, considerable uncertainty regarding the values of the model’s parameters hinders a comprehensive empirical evaluation of its prediction. For these reasons we provide general policy implications on p. 17. Yet, following the suggestion of R1, we have added a new paragraph (at the bottom of p. 17, and the top of p. 18), presents a preliminary evaluation of the model, based on rough estimates of our model parameters in this context of the COVID-19 pandemic, and existing donation data. These data offer preliminary and tentative credence to the model predictions.

Reviewer 2

We sincerely thank the Reviewer for carefully reading our Manuscript and for the helpful comments. Below please find the Reviewer's comment (black fonts) followed by our response (blue & bold fonts).

This work concerns a specific situation around 'vaccine-diplomacy' from the game theoretical standpoint, which seems new, no one has ever explored. In this sense, I can positively evaluate this work.

A vaccine-rich country takes strategy $s=\{0,1,2\}$ implying the amount of surplus doses she stored to donate vaccine-poor countries. This is not simply donation; just wasting her money, because if poor countries fail to supply vaccine, the probability of a variant strain emerging increases, which also impacts (may devastates) rich countries. Thus, as a whole, the situation encompasses a social dilemma, which might be same as the so-called public-goods game-type dilemma, i.e., either Defector ($s=0$ is D-strategy) – dominate (like Prisoner's Dilemma) or Polymorphic (co-existence of C and D; like Chicken game). Such a story seems somehow attracting to the audience.

We thank the Reviewer for the nice summary of our model and for emphasizing the novelty of our work.

The result they explored visually suggests optimal state along the parameter of variant emerging (λ) and coverage of vaccination (v_{max}). I'm bit confused whether why the authors did not simply show Nash Equilibrium (NE) besides those social optimal (SO) state. If there is a gap between NE and SO visually, it is literally working as a hard evidence of existence of social dilemma. Some previous study gives theoretical framework of this gap to measure 'social dilemma', called Social Efficiency Deficit (SED). I strongly suggest the authors to extend their visual result to such discussion by referring to the concept of SED and relevant literatures (for example; (i) Social efficiency deficit deciphers social dilemmas, Scientific Reports 10, 16092, 2020).

The Reviewer suggests that we demonstrate how the optimal solution differs from the stable solution in those parameter regions where they are not identical.

Accordingly, in the revised Manuscript, we added a figure (Fig. 3) showing the differences between the stable and the optimal solution:

In particular, if the optimal solution is stable, then all countries contribute the optimal amount, and this corresponds to the white regions in Fig. 3. However, in the colored regions, the optimal solution is unstable. Panel A shows the number of countries that defect and do not contribute the optimal amount in the stable solution. Panel B shows the difference between the cost following the optimal solution and the cost following the stable solution.

We also added the following to the main text: “Finally, note that in other regions, donation of two doses (region 2) or one dose (region 5) is optimal (maximizes the social welfare of the vaccine-rich countries as a whole), but the optimal solution is unstable. Instead, the stable solutions in these parameter regions are sub-optimal and dictate donation by only some of the countries. Fig. 3 shows the number of countries that do not donate following the SEA solution (Fig. 3A), and the corresponding “social efficiency deficit” or “cost of anarchy”—shows the difference between the cost following the optimal solution and the cost following the SEA solution (Fig. 3B)” (pages 14-15).

When it comes to the term ‘vaccination game’ as one of the jargons is the merged framework

of mathematical epidemiology, quantifying disease spreading, with evolutionary game theory, where the dynamics of each agent's decision, e.g., whether he/ she committing a vaccination or not, is fairly quantified. Such previous studies are different from what the authors contributing in the present study. And the authors should introduce those by citing several precursors. There are several review papers and books; e.g, (ii) Sociophysics Approach to Epidemics, Springer, 2021.

We are grateful for these important and relevant suggestions. In the revised manuscript (page 5) we refer to these studies, and explain that they address individual-level decisions to vaccinate or not, assuming the availability of vaccines for the studied population. Our study, on the other hand points to the significance of international vaccine inequality in the context of a pandemic, notably the fact that in many countries vaccines are in short supply or practically unavailable, while other countries possess surplus stocks of vaccines. These surplus stocks are controlled by the governments of these vaccine-rich countries, and these country-level decisions are the focus of our model.

REVIEWERS' COMMENTS:

Reviewer #1 (Remarks to the Author):

The manuscript has significantly improved. All my previous comments have been addressed. I am satisfied with the manuscript in its current form and suggest it is accepted.

Reviewer #2 (Remarks to the Author):

The revised MS is adequate to be published at the journal. Thus, I'm pleased to suggest acceptance.